# Monitoring Subsidence Area with the Use of Satellite Radar Images and Deep Transfer Learning

**DOI:** 10.3390/s22207931

**Published:** 2022-10-18

**Authors:** Anna Franczyk, Justyna Bała, Maciej Dwornik

**Affiliations:** Department of Geoinformatics and Applied Computer Science, AGH University of Science and Technology, al. Mickiewicza 30, 30-059 Kraków, Poland

**Keywords:** subsidence detection, neural network, image analysis

## Abstract

Subsidence, especially in populated areas, is becoming a threat to human life and property. Monitoring and analyzing the effects of subsidence over large areas using in situ measurements is difficult and depends on the size of the subsidence area and its location. It is also time-consuming and costly. A far better solution that has been used in recent years is Differential Interferometry Synthetic Aperture Radar (DInSAR) monitoring. It allows the monitoring of land deformations in large areas with high accuracy and very good spatial and temporal resolution. However, the analysis of SAR images is time-consuming and involves an expert who can easily overlook certain details. Therefore, it is essential, especially in the case of early warning systems, to prepare tools capable of identifying and monitoring subsidence in interferograms. This article presents a study on automated detection and monitoring of subsidence troughs using deep-transfer learning. The area studied is the Upper Silesian Coal Basin (southern Poland). Marked by intensive coal mining, it is particularly prone to subsidence of various types. Additionally, the results of trough detection obtained with the use of convolutional neural networks were compared with the results obtained with the Hough transform and the circlet transform.

## 1. Introduction

Like any other detection technique, differential interferometry synthetic aperture radar (DInSAR) has attracted considerable attention in the field of subsidence monitoring. It provides information on ground deformation [1,2,3,4,5] including mining-related subsidence. It is particularly troublesome for the Upper Silesian Coal Basin (USCB) (southern Poland). The USCB is one of the largest coal basins in Europe, where mining activity started as early as the 18th century and has continued to this day. Due to multi-seam coal mining, vertical displacements in the USCB reach in some places up to dozens of meters. It is estimated that they affect over 300 km ${}^{2}$ of land, and even if it is mainly farmlands and forests, ground deformations periodically occur in urban areas contributing to the destruction of buildings and infrastructure. Within the next 30 years, all mines are planned to close down as part of the transformation plan of the mining industry in Poland [6].

Monitoring of mining-related deformations is imposed by law and is used to verify subsidence forecasts, maximize coal extraction, and minimize hazard to people, infrastructure, and the environment. Subsidence monitoring that applies traditional techniques, such as in situ measurements using total stations or global navigation satellite systems (GNSS), is laborious and time-consuming, especially when the areas investigated grow in size over time [7,8]. Therefore, monitoring is only used on selected areas of the size of a few square kilometers [8]. Mining measurements on level lines are carried out once a year and sometimes even less frequently [9]. Moreover, those are performed point by point, so the spatial extent is insufficient to assess the subsidence mechanism [8,10]. A much better solution applied in recent years is the monitoring using DInSAR [11,12,13].

The differential interferometry synthetic aperture radar technique consists in processing two SAR images of the same part of the Earth. The phase difference of the returning signal of two images of the same area taken at different times is used to detect ground deformations [14,15]. The DInSAR data processing method yields an image called a differential interferogram that shows the movement of the ground that occurs between the measurements with a centimeter accuracy and a decametric resolution. The displacement is calculated by subtracting the phase component of the two SAR images of the same area. Deformations are visible in the interferogram as interferometric fringes, and the density of the fringes is correlated with the size of the vertical displacement.

A significant advantage of this method is that it is possible to regularly monitor large areas, even up to thousands of square kilometers. The data is provided because of the international space missions, including the Copernicus program. The Copernicus program is initiated and financed by the European Union [16]. The major problem in the case of such a large and regular amount of data provided by the satellites is to search for a differential interferogram and find interferometric fringes that may indicate subsidence. This task is particularly difficult due to the deformation of the fringes and the noise, which are connected with land cover changes (vegetation, snow cover, etc.) or atmospheric disturbances. Atmospheric patterns, which can be similar to the trough patterns, can not be distinguished from real subsidence on one differential interferogram. This problem can be reduced by utilizing the relationship between elevation and atmosphere [17], statistical analysis [18], or using external data [19].

Many methods are proposed to detect subsidence troughs in DInSAR images: Gabor transformation [20], Hough transformation (HT) [21], template recognition [22], Convolutional Neural Networks (CNN) [23], Circlet Transform (CT) [24], or slope analysis [25]. Each of these methods has its limitations and different efficiency in trough detection in noisy satellite images. Most of them can be used to support the detection of subsidence in larger SAR images, but they are not suitable for fully automated operations [25]. Additionally, due to the gear requirements and calculation cost, their application for constant monitoring is highly limited.

In this paper, we propose the use of deep transfer learning for detection and monitoring of subsidence.

The literature contains many examples of the application of neural networks [26,27]. Deep convolutional networks are used in Earth sciences and remote sensing. The data available for training is usually scarce compared to the requirements of deep networks. One of the ideas on how to overcome the problem of network retraining is to limit the neural connections [28] or to generate a large amount of artificial training data [29]. The use of convolutional neural networks (CNN) to detect subsidence in interferometric images was presented in [23]. The solution proposed there is time-consuming and requires a specialized computing environment. Like in the case of the methods mentioned above, its efficiency in subsidence detection in interferograms is not 100% either. CNNs were introduced in the late 1980s [30] and improved through the years. During this, networks such as ZFNet [31], GoogleNet [32], VGG [33], ResNet [29], MobileNet [34], and SENet [35] were proposed and used for classification task. In the problem of detection of subsidence troughs, a system based on a one-stage method, namely, you look only once [36,37,38] architecture, was developed [23].

In this paper, the authors propose the use of re-trained residual convolutional network for a fast, almost instantaneous classification of potential subsidence in differential interferograms. The results obtained in this way can subsequently be used for detection of troughs, their detailed analysis, as well as for monitoring and evaluation of subsidence prone areas by early warning systems or an expert.

The deep transfer learning method using ResNet-50 [39] is used to detect the interferogram area where a subsidence trough may exist. The pre-trained image classification network that has already learned to extract informative features from natural images was used as a starting point. The image of the troughs on the interferograms is significantly different from the images that were used during the training of the residual network; therefore, the net was fine-tuned so it was able to learn features specific to a new data set.

The novelty of the research is the customization of ResNet-50 to the problem of detection of subsidence troughs, as well as the performance analysis of the proposed solution. Experiments involving replacing ResNet with AlexNet considerably worsened the detection rate (so the results are not included in this letter). Experiments with fine tuning of last few layers did not improve the detection results due to the different appearance of fringe patterns. Fringe patterns are more difficult to define than typical objects classified for automatic target recognition (e.g., aircraft, ships, etc.).

## 2. Materials and Methods

As it is difficult to determine the set of parameters that identify troughs, the problem of its detection is complex. Troughs visible in interferograms do not have regular shapes. They are usually elliptical or circular. Subsidence areas can manifest themselves in the form of characteristic fringes, but their number, shape, and occurrence can vary. An additional problem is high noise in interferograms. It can look like a fringe distribution typical of subsidence, but it can also obscure a trough. Therefore, interferograms often show only a fragment of a trough and the typical fringes are blurred.

An important issue in trough detection is the small number of training samples, which poses a problem for many machine learning algorithms, especially for methods based on deep convolutional neural networks. In this paper, we propose a trough detection algorithm divided into two parts: training (Figure 1) and detection (Figure 2). In the first stage of the works, a database of trough images visible in interferograms was constructed. It involved 23 interferograms computed from images taken by Sentinel-1 in the period of time from September 2017 to January 2018 in the USCB (Figure 3).

Sentinel-1 is a polar orbiting radar imaging system consisting of two satellites (Sentinel-1A and Sentinel-1B) in the EU Sentinels constellation. A differential interferogram was computed on the basis of SAR image pairs acquired by the C-band satellite (wavelength 5.6 cm) along the descending orbit at an interval of twelve days but with the same illumination geometry. The interferogram was generated using SNAP software and filtered using the Goldstein filtering method. The analyzed interferograms were monochromatic with 256 levels of gray intensity. Conversion from double to unsigned integer type significantly reduced calculation time without adversely affecting the accuracy of the analysis. The troughs visible on the set of 23 interferograms were classified manually and independently by three operators. The database of the trough images was created only for those interferogram fragments in which all the operators classified as subsidence areas. Due to the nature of the troughs in the USCB, the trough images that fed the database were 100 × 100 px, which correspond to an area about 1100 × 1100 m. Then, this set of 313 troughs was enhanced using common methods in machine learning [40]. The enhancements applied were based on four operations: negation (255−image), 90-degree rotation, Gaussian mask average filtering (5 × 5, std = 1), and scaling (squeezing in horizontal axis up to 72% and supplementing with uniform noise at the edges). It increased the size of the data set to 1571 images with visible subsidence troughs. Examples of the enhanced operations are presented in Figure 4.

The data set had the following features:All images were of the same size, selected based on the nature of the subsidence occurring in the analyzed area;Each image showed only one subsidence trough;Of the analyzed set 23 interferograms, a negative set was created that contains fragments of the interferogram with no visible subsidence troughs;Specified proportions were kept between the troughs containing the set and the interferogram fragments without a visible subsidence area;In the cross-validation stage, the ratio of the learning set to the test set was 70/30.

The deep transfer learning method with residual network (ResNet) has been proposed to tackle the subsidence detection on SAR interferograms. Residual networks, unlike traditional sequential neural networks, have “shortcut connections” modules. These modules were introduced into the residual network to skip one or more layers and perform identity mapping. Identity mapping does not have any parameters and is only there to add the output from the previous layer to the layer ahead. The application of a shortcut connection allows one to train much deeper networks than what was possible using traditional sequential neural networks.

ResNet-50, used in the detection of subsidence throws and shown in Figure 5, contains four residuals blocks, repeated 3, 4, 6, and 3 times, respectively. ResNet-50 has three convolution layers in each block, followed by average pooling and a fully connected layer related to the number of class detected. In the experiment described in this paper, there were only two classes differentiating the analyzed images as an area with subsidence trough or an area without subsidence.

ResNet-50 pre-trained on ImageNet images [29] was applied as our initial model. However, the large difference lying between interferogram patches and natural object images that were used to train the network, harms direct transferability. The residual network trained on colored images was modified to be able to classify monochrome images. It was achieved by changing the dimension of the first convolutional layer of the pre-trained network and setting its weight as a sum of all pre-trained RGB channels weights. This technique is commonly used in the transfer learning of monochromatic images. The network was trained on a new data set, described in the previous section, with the pre-trained weight as a starting point. It was made possible to the learn features specific to the data set obtained from interferograms.

## 3. Results

ResNet-50 was trained using 1571 labeled subsidence troughs and 1421 images received from inteferograms without visible traces of subsidence. The size of the input patches has been scaled to the size of 224 × 224 px. We used the stochastic gradient descent (SGD) method for training, with a minibatch size of 10, a momentum of 0.9, and initial learning rate equal to 3 × 10${}^{-4}$. The training included 1088 iterations and lasted half an hour on a personal computer. The accuracy and loss function are shown in Figure 6. Accuracy, used as a statistical measure of how well a binary classification test correctly identifies or excludes a condition, is the proportion of correct predictions (both true positives and true negatives) among the total number of cases examined [41].
(1)$$\mathrm{Accuracy}=\frac{TP+TN}{TP+TN+FP+FN}$$
where *TP* (True positive)—detection of existing troughs; *FP* (False positive)—false detection of the subsidence area; *TN* (True negative)—correct indication of the area without subsidence; *FN* (False negative)—undetection of the existing subsidence area.

The confusion matrix and confusion matrix–based performance metrics used to assess the significant improvement of the proposed classification model are presented in Figure 7.

The trained neural network was used for automatic classification of areas of subsidence on the interferogram calculated for the study area from satellite images taken on the days of 10 and 22 October 2016. Two areas of 15.4 km × 22 km with a larger cluster of subsidence troughs were chosen for the test of proposed detection algorithm. Red circles indicate that the visible subsidence basins were created by a human annotator (Figure 8 and Figure 9).

The classification was made for images of a constant size of 100 × 100 px taken from the interferogram with a spacing equal to 5 px. As in the retraining process, before classification, a images were scaled to the size of 224 × 244 px, required by the ResNet-50 architecture. The data set obtained in this way from the interferogram consists of 98,800 images. Classification with the use of a trained neural network took a quarter of an hour. In Figure 10 and Figure 11 we present images that were classified as the subsidence trough.

The detection results obtained with the retrained ResNet-50 network for the interferograms presented in Figure 8 and Figure 9 were compared with the detection results carried out using the Hough transform (HT [21]) and the circlet transform (CT [24]). The following parameter values were adopted for the circlet transform: N = 5, threshold = 1.1, and the range of the searched rays is from 20 to 60 pixels. In the case of the Hough transform, Canny high-pass filtering was used for edge detection with a threshold of 0.5, and the radius range was 20 to 80 pixels.

The referenced indications of the subsidence areas (red circles in Figure 8 and Figure 9) were carried out by human annotator on the basis of the patterns visible in radar image as well as his experience and knowledge of the analyzed area. The classification algorithm detected all subsidence areas indicated by the annotator in the first interferogram (Figure 10). However, there is one false positive detection—a misclassified area without subsidence troughs. The false positive detection was caused by the occurrence of patterns, which confusingly resembled the image of a subsidence trough on the interferogram. The area was not indicated by an experienced annotator familiar with the research area.

The classification results obtained for the second interferogram are slightly worse. The detection algorithm correctly indicated 7 of 8 existing subsidence areas. Moreover, four areas without subsidence were incorrectly indicated as troughs. Among all 11 indicated subsidence areas, it is possible to exclude those areas with a small amount of indication. The detection of subsidence areas was carried out for images obtained from the interferogram with an interval of 5 pixels. For this reason, areas of subsidence indicated only several times (2, 4, and 5 times for the second interferogram) may be excluded as unreliable indications. Among the remaining indications, there is still one false positive and one false negative classification. False positive classification indicated an area of subsidence, which is formed by randomly distributed patterns deceptively similar to the image of the trough in the interferogram. Finally, in false negative classification, the area of subsidence not detected by the algorithm was caused by a blurry and poorly visible pattern.

The detection results obtained with the use of transfer learning technique were compared with the subsidence area detection obtained with the use of the HT and CT method (Figure 12, Figure 13, Figure 14 and Figure 15). The best detectability was obtained with the use of the deep transfer learning method (18 detected and 2 incorrectly indicated, 1 undetected) for 19 existing subsidence areas. The other two methods were less effective in trough detection (14 indications for CT and 10 for HT). Additionally, CT indicated 8 and HT indicated 6 areas incorrectly, so-called false indication. False indications were related to noise in the interferogram, which in this area had a spherical shape similar to the shape of a trough (Figure 12).

The calculation time of the re-trained ResNet, CT, and HT is shown in Figure 16b. The results were archived using personal computer (Intel Core i7 8gen., 32 GB RAM, SSD hard disk, Windows 10). ResNet has proven to be the slowest detection algorithm. The ResNet computation time, when using an already learned network, is almost 4 times longer compared to both the circlet and Hough transforms. The computation time for CNN and CT did not depend on the distribution of the pixel values in the image. In the case of the Hough transform, the computational time depended on the level of noise that generated false edges.

## 4. Conclusions

The aim of the study was to develop a deep transfer learning method for the problem of detection and monitoring of subsidence in SAR interferograms. The main problem with detecting troughs compared to normal object detection tasks is the fact that their image may be poorly visible on interferograms. Moreover, some subsidence images are blurred or noisy. All these factors make it difficult for both experts and automated detection systems to identify subsidence prone areas and to monitor them. Rather than applying the ResNet-50 pretrained model to interferogram images directly, we use images with visible subsidence trough and images where there are no features characteristic for land subsidence to fine-tune the residual network. The achieved results made it possible to correctly classify all subsidence areas visible in the interferogram of the selected part of the USCB area. Results obtained with convolutional neural networks, for tested interferograms, are more reliable than circlet or Hough transform.

## Figures and Tables

**Figure 1 sensors-22-07931-f001:**
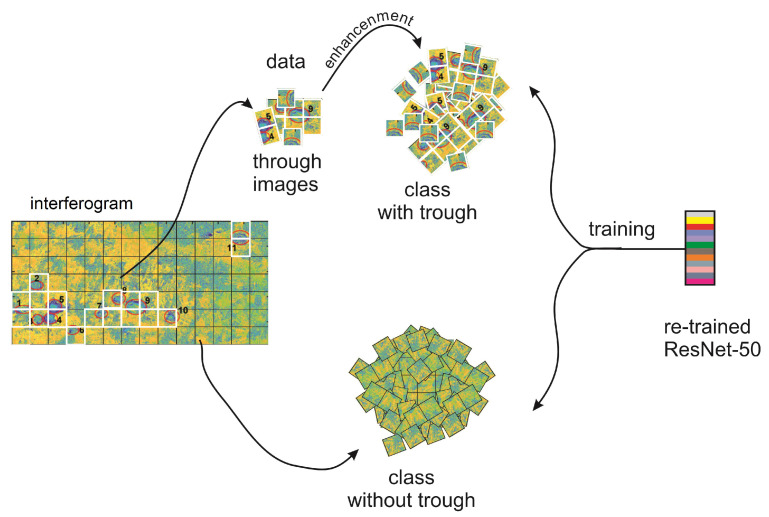
Training stage used in the presented method.

**Figure 2 sensors-22-07931-f002:**
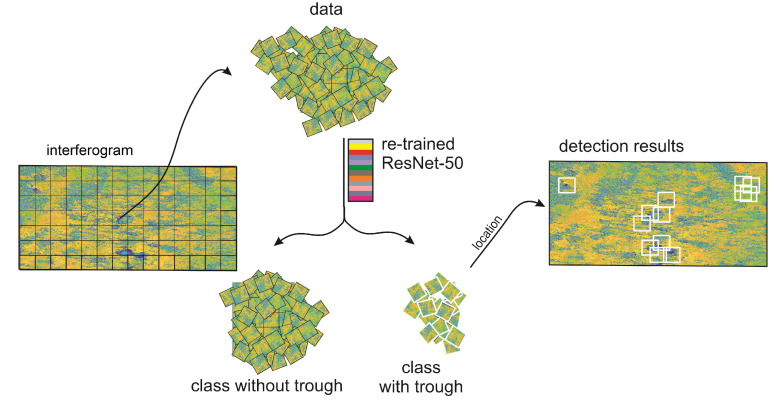
Detection stage used in the presented method.

**Figure 3 sensors-22-07931-f003:**
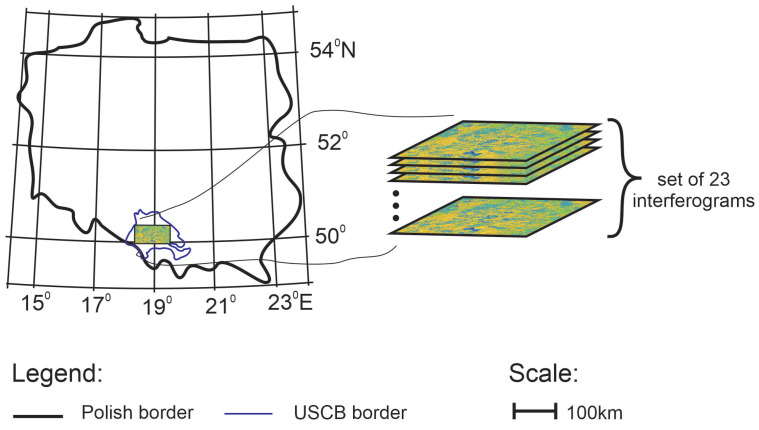
A set 23 differential interferograms generated for recordings performed from September 2017 to January 2018. It was calculated from radar images of the Upper Silesia Coal Basin area, Poland.

**Figure 4 sensors-22-07931-f004:**
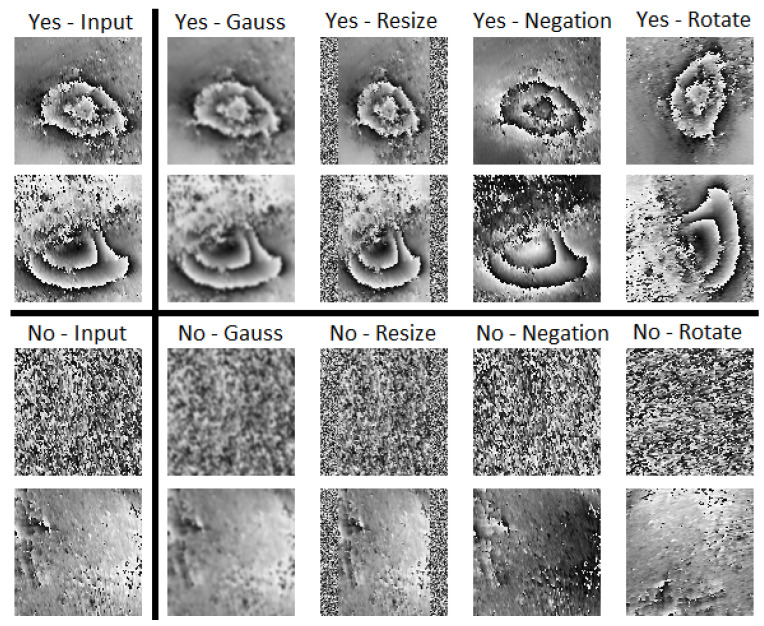
Images comprising the test set after data augmentation.

**Figure 5 sensors-22-07931-f005:**
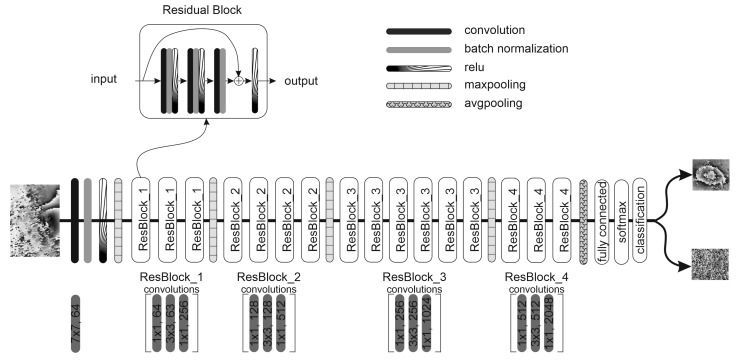
Residual network architecture applied in our experiments.

**Figure 6 sensors-22-07931-f006:**
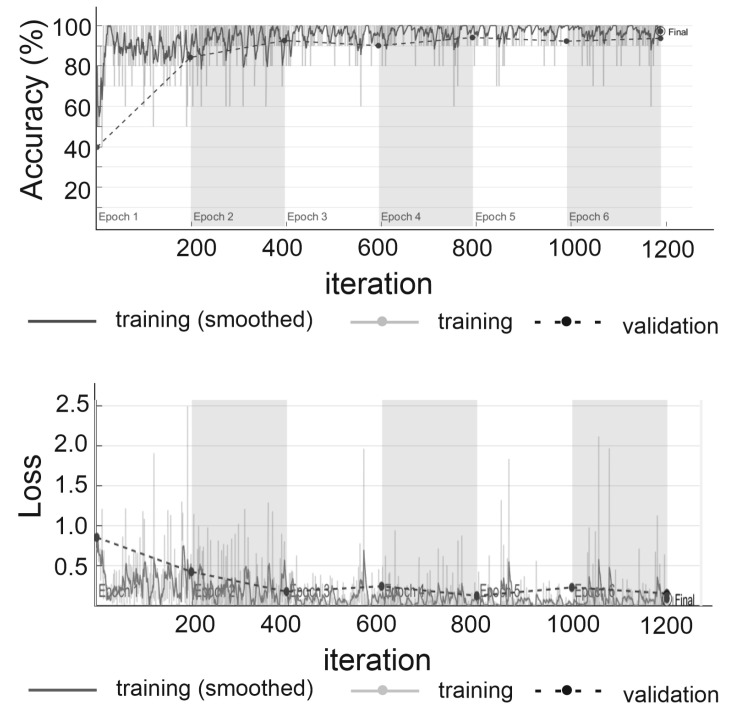
Accuracy and loss function variability depending on the number of iterations obtained during ResNet-50 training using the interferometric data set.

**Figure 7 sensors-22-07931-f007:**
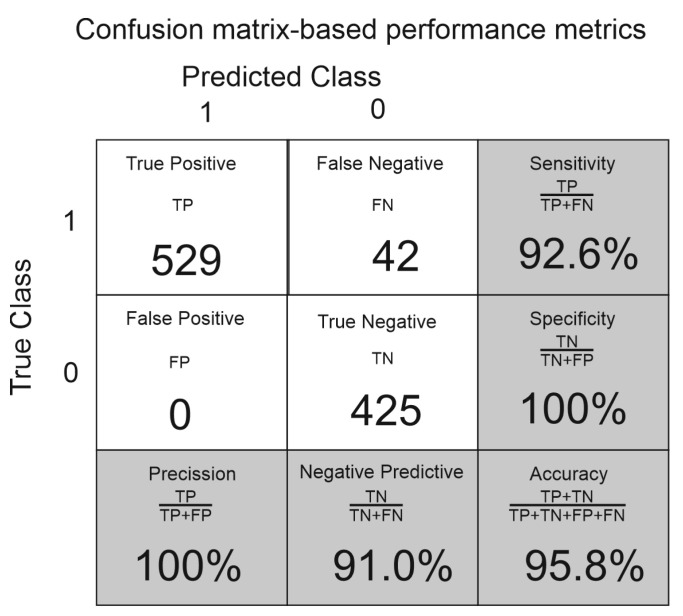
Confusion matrix and performance metrics of the validation data set. The subsidence area is labeled is “1”, while “0” describes areas without subsidence.

**Figure 8 sensors-22-07931-f008:**
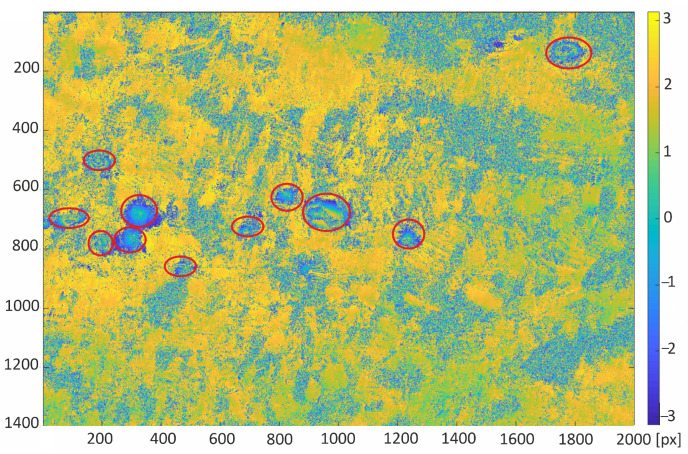
Differential interferogram of the first area with marked subsidence areas (red ellipses) calculated for the study area from satellite images taken on the 10th and 22nd of October 2016.

**Figure 9 sensors-22-07931-f009:**
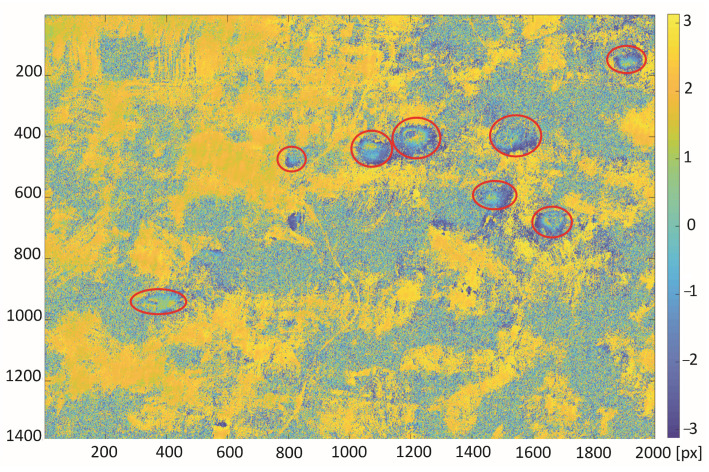
Differential interferogram of the second area with marked subsidence areas (red ellipses) calculated for the study area from satellite images taken on the 10th and 22nd of October 2016.

**Figure 10 sensors-22-07931-f010:**
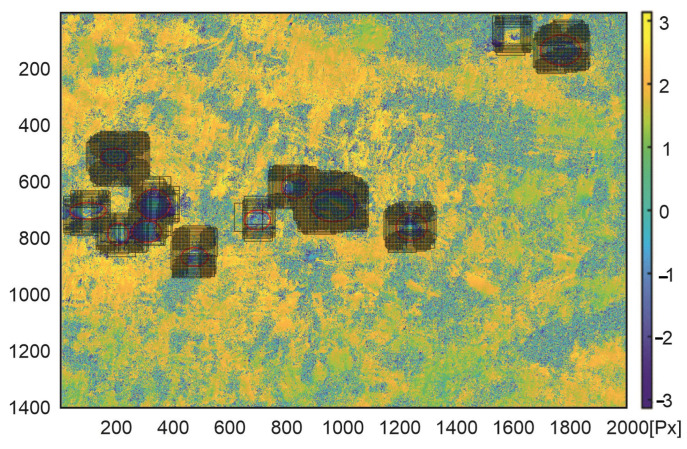
Interferogram of the first area computed from pictures recorded by Sentinel-1 on the 10th and 22nd of October 2016. Red ellipses are the subsidence area marked by experts; black squares were detected by the re-trained ResNet-50 network.

**Figure 11 sensors-22-07931-f011:**
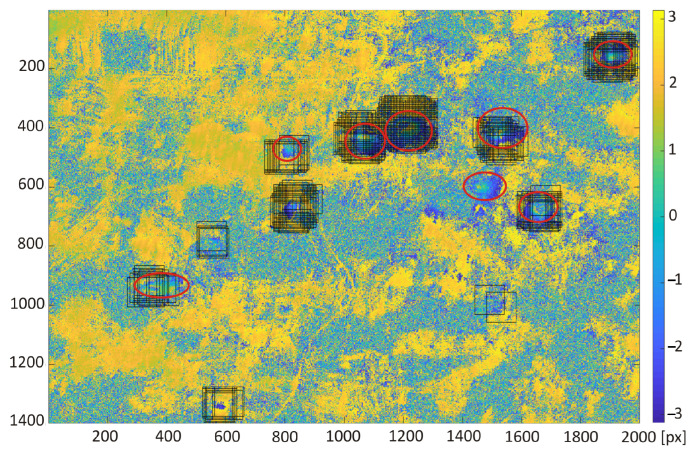
Interferogram of the second area computed from pictures recorded by Sentinel-1 on the 10th and 22nd of October 2016. Red ellipses are the subsidence area marked by experts; black squares were detected by the re-trained ResNet-50 network.

**Figure 12 sensors-22-07931-f012:**
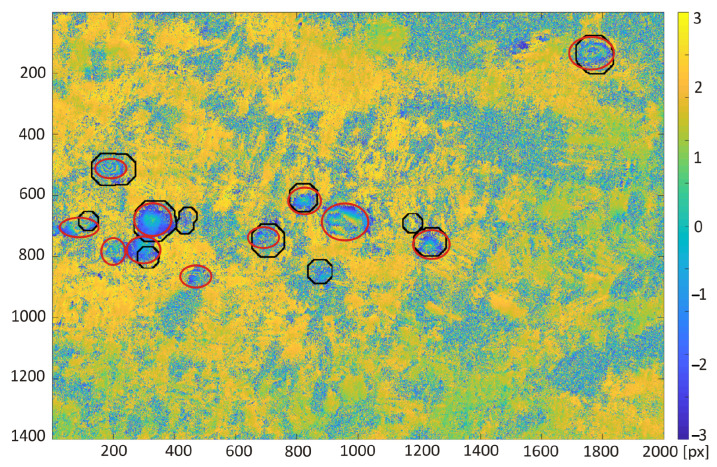
Interferogram of the first area computed from pictures recorded by Sentinel-1 on the 10th and 22nd of October 2016. Red ellipses are the subsidence area marked by experts; black lines were detected by circlet transform.

**Figure 13 sensors-22-07931-f013:**
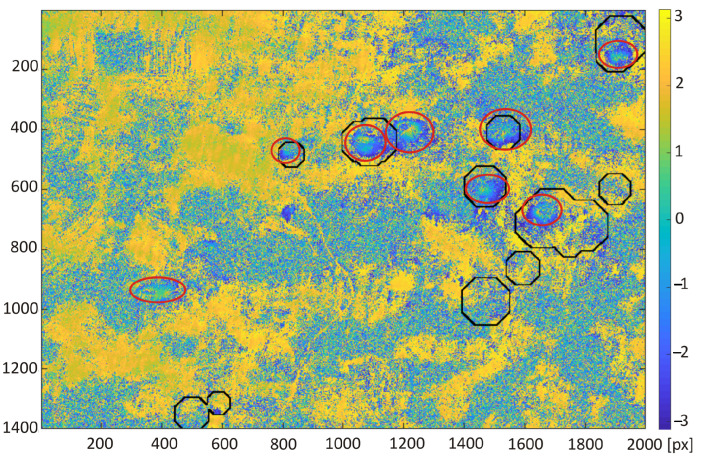
Interferogram of the second area computed from pictures recorded by Sentinel-1 on the 10th and 22nd of October 2016. Red ellipses are the subsidence area marked by experts; black lines were detected by circlet transform.

**Figure 14 sensors-22-07931-f014:**
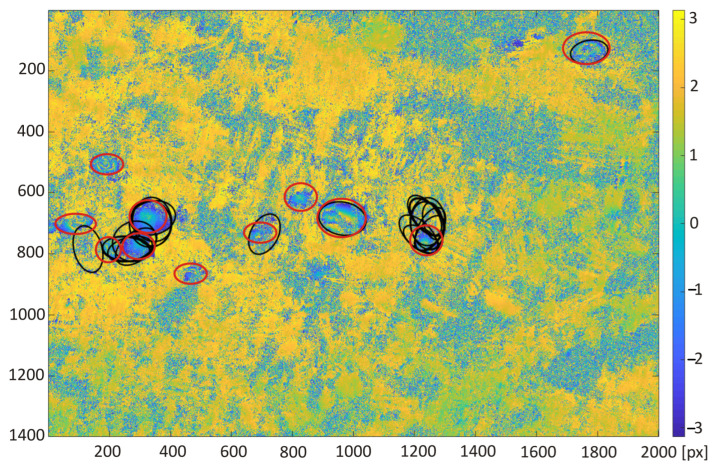
Interferogram of the first area computed from pictures recorded by Sentinel-1 on the 10th and 22nd of October 2016. Red ellipses are the subsidence area marked by experts; black ellipses were detected by Hough transform.

**Figure 15 sensors-22-07931-f015:**
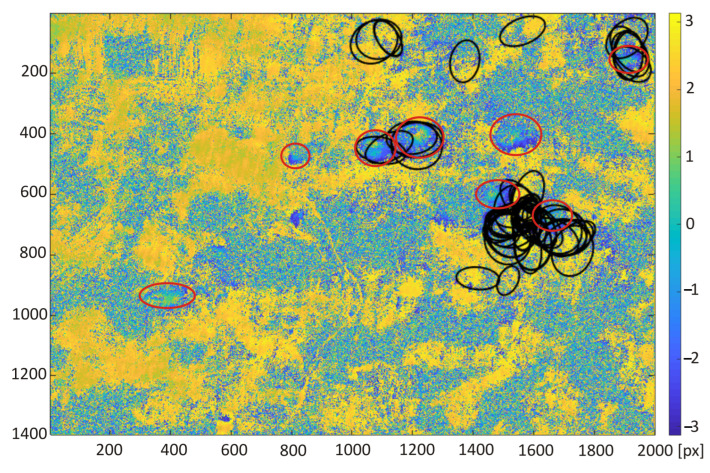
Interferogram of the second area computed from pictures recorded by Sentinel-1 on the 10th and 22nd of October 2016. Red ellipses are the subsidence area marked by experts; black ellipses were detected by Hough transform.

**Figure 16 sensors-22-07931-f016:**
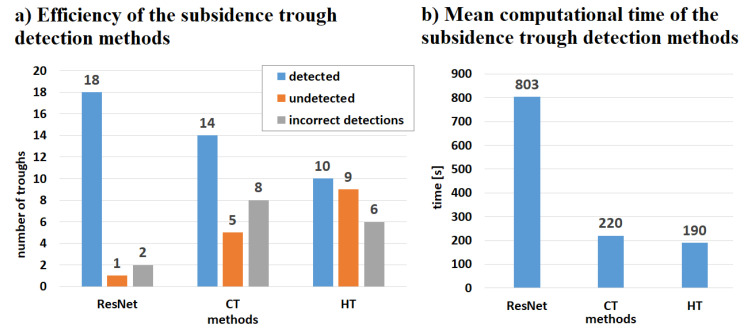
Efficiency (**a**) and computational time (**b**) measured for the tested interferograms (Figure 8 and Figure 9). The computational time of the ResNet-50 network image classification was measured only for the classification process.

## Data Availability

The datasets used and analyzed during the current study are available from the corresponding author on request.

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
