# Peer review of "Monitoring Subsidence Area with the Use of Satellite Radar Images and Deep Transfer Learning"

_sensors, 2022, doi:10.3390/s22207931_

Round 1

Reviewer 1 Report

line 32, space between the sentence and references 6,7

lines 49-50 provide reference

lines 61-63 provide reference or elaboration

line 71 provide reference

The caption of figure 1, "A set of 23"

Either mention pixels or define px in the image size. 

lines 121,123,125; improper sentences that hinder understanding. 

line 116, how were the images enhanced, need more details and elaboration

Section 2 needs better organization through subsections. 

Cite the ResNet paper.

Figure 6, what are the two classification classes?

line 154 typo "through" --> trough

Details of the dataset on the number of images are unclear, once it is mentioned as 313 and later different numbers.

The number of epochs rather than iterations should be mentioned.

The detailed specifications of the computing platfrom should be mentioned in order to appreciate the time reported. 

How is the output of the ResNet classification model used for determining the location?

Define the accuracy. 

List of abbreviations is missing. 

Author Response

We would like to thank the Reviewers for both their time and valuable comments on the article.  Thorough analysis of the article carried out by all Reviewers helped us to correct and improve our work. The answers to the Reviewers comments are listed below.

Reviewer #1

  1. line 32, space between the sentence and references 6,7
  2. lines 49-50 provide reference
  3. lines 61-63 provide reference or elaboration
  4. line 71 provide reference
  5. The caption of figure 1, "A set of 23"
  6. Either mention pixels or define px in the image size. 
  7. lines 121,123,125; improper sentences that hinder understanding. 
  8. line 116, how were the images enhanced, need more details and elaboration
  9. Section 2 needs better organization through subsections. 
  10. Cite the ResNet paper.
  11. Figure 6, what are the two classification classes?
  12. line 154 typo "through" --> trough
  13. Details of the dataset on the number of images are unclear, once it is mentioned as 313 and later different numbers.
  14. The number of epochs rather than iterations should be mentioned.
  15. The detailed specifications of the computing platfrom should be mentioned in order to appreciate the time reported. 
  16. How is the output of the ResNet classification model used for determining the location?
  17. Define the accuracy. 
  18. List of abbreviations is missing. 

Typos, missing references and grammatical issues mentioned as detailed comments 1-5, 10,12 were corrected.

Definition of image size was introduced (line 127).

Paragraph with lines 121, 123 and 125 was rewritten (lines 133-142).

Image enhancement was described in detailed (lines 129-131) and included in the description of the detection algorithm in Figure 1.

Section 2 was reorganized by adding description of the detection algorithm in Figure 1 and 2 and replacing performance result to the next section.

A class definition has been added (Figure 8 caption)

Information of the number of the images in dataset was put in order (line 127-131)

The detailed specification of the computing platform was added (line 229-230).

The illustration of the classification algorithm was presented on Figure 1 and 2.

The definition of accuracy was added to article (Equation 1, lines 177-179).

List of aberration was wadded to article (line 257).

Reviewer 2 Report

Following comments are needed to update in the paper:

1.      Need to include novelty part in the abstract that can show the need of the study

2.      Line 20—the words ‘problem’ and ‘the’ must be removed.

3.      Line 22—istead of ‘continue to date’ write ‘continue till date’.

4.      Line 26—please mention the reference for ‘Within the next 30 years, all mines are to be closed down as part of the transformation plan of the mining industry in Poland.’

1.      Line 31 -- write full form of GNSS

2.      Line 32— rewrite the sentence without using ‘This is why’

3.      Line 35—instead of ‘ they’ please do write ‘those’

4.      Line 48—remove ‘of’

5.      Line 49—rewrite sentence   The data is provided thanks to the international space missions including those that are part of the Copernicus program, initiated and financed by the European Union.’

6.      Line 55—write ‘distinguished’ instead of ‘ distinguish’

7.      Line 56—write ‘can be reduced’ instead of ‘could be reduce’

8.      Line 59—write ‘many’ instead of ‘A lot of’

9.      Line 61—write ‘these’ instead of ‘those’

10.   Line 62—write ‘therefore’ instead of ‘which is why’

11.   Line 63—write ‘ these’ instead of ‘them’

12.   Line 64 &65—rewrite the sentence for better understanding ‘What it more, due to the gear requirements and the calculation cost their application for constant monitoring is highly limited.’

13.   Line 88—remove the word  ‘your’

14.   There is a need for enhancement of the state-of-the-art and also include the novelty of the study in the 4-5 bullet points, which can show the advantage and reason of the study.

15.   Kindly include a proposed approach in form of a graphical form so that a new comer/new researcher may understand whole work done and procedure in this paper

16.   Line 92—write ‘ its’ instead of ‘their’

17.   Line 96—write ‘therefore’ instead of ‘It is for that reason that’

18.   Line 121—write ‘contains’ instead of ‘contains’

19.   Line 128—put comma after ‘in the cross-validation stage’

20.   Deep transfer learning method has several disadvantages such as  1) requires a very large, general dataset, 2) hard to show consistent generic gains, 3) negative transfer, 4) whenever user use transfer learning, user’s training data should have two options, 5) User can not remove layers with confidence to reduce the number of parameters, etc. In such conditions, how the proposed approach manages the all disadvantages? Kindly include the validation and proof for all solutions.

21.   residual network (ResNet) usually requires weeks for training, making it practically infeasible in real-world applications. In this paper how did the author manage this type of problem? Kindly include the validation for this

22.   Line 132—remove ‘so called’

23.   Line 158—write the number ‘1 088’ without middle space after 1

24.   Kindly extent the explanation of Fig. 7-10 to make it more understandable

25.   Line 171—write the number ‘98 800 ‘ without middle space after  8

26.   Line 173—correct the spelling ‘through’

27.   Line 178—write ‘rays are from’ instead of ‘rays from’

28.   Line 187—write ‘were’ instead of ‘was’

29.   Kindly include the value of each bar in the figure 11

30.   Line 203—write correctly the word ‘through’ (also please check the complete manuscript for this word and correct accordingly)

31.   Line 207—write ‘are’ instead of ‘were’

32.   Line 213—remove the word ‘reasonable’

Author Response

We would like to thank the Reviewers for both their time and valuable comments on the article.  Thorough analysis of the article carried out by all Reviewers helped us to correct and improve our work. The answers to the Reviewers comments are listed below.

Reviewer #2

Following comments are needed to update in the paper:

  1. Need to include novelty part in the abstract that can show the need of the study
  2. Line 20—the words ‘problem’ and ‘the’ must be removed.
  3. Line 22—istead of ‘continue to date’ write ‘continue till date’.
  4. Line 26—please mention the reference for ‘Within the next 30 years, all mines are to be closed down as part of the transformation plan of the mining industry in Poland.’
  5. Line 31 -- write full form of GNSS
  6. Line 32— rewrite the sentence without using ‘This is why’
  7. Line 35—instead of ‘ they’ please do write ‘those’
  8. Line 48—remove ‘of’
  9. Line 49—rewrite sentence   ‘The data is provided thanks to the international space missions including those that are part of the Copernicus program, initiated and financed by the European Union.’
  10. Line 55—write ‘distinguished’ instead of ‘ distinguish’
  11. Line 56—write ‘can be reduced’ instead of ‘could be reduce’
  12. Line 59—write ‘many’ instead of ‘A lot of’
  13. Line 61—write ‘these’ instead of ‘those’
  14. Line 62—write ‘therefore’ instead of ‘which is why’
  15. Line 63—write ‘ these’ instead of ‘them’
  16. Line 64 &65—rewrite the sentence for better understanding ‘What it more, due to the gear requirements and the calculation cost their application for constant monitoring is highly limited.’
  17. Line 88—remove the word  ‘your’
  18. There is a need for enhancement of the state-of-the-art and also include the novelty of the study in the 4-5 bullet points, which can show the advantage and reason of the study.
  19. Kindly include a proposed approach in form of a graphical form so that a new comer/new researcher may understand whole work done and procedure in this paper
  20. Line 92—write ‘ its’ instead of ‘their’
  21. Line 96—write ‘therefore’ instead of ‘It is for that reason that’
  22. Line 121—write ‘contains’ instead of ‘contains’
  23. Line 128—put comma after ‘in the cross-validation stage’
  24. Deep transfer learning method has several disadvantages such as  1) requires a very large, general dataset, 2) hard to show consistent generic gains, 3) negative transfer, 4) whenever user use transfer learning, user’s training data should have two options, 5) User can not remove layers with confidence to reduce the number of parameters, etc. In such conditions, how the proposed approach manages the all disadvantages? Kindly include the validation and proof for all solutions.
  25. Residual network (ResNet) usually requires weeks for training, making it practically infeasible in real-world applications. In this paper how did the author manage this type of problem? Kindly include the validation for this
  26. Line 132—remove ‘so called’
  27. Line 158—write the number ‘1 088’ without middle space after 1
  28. Kindly extent the explanation of Fig. 7-10 to make it more understandable
  29. Line 171—write the number ‘98 800 ‘ without middle space after  8
  30. Line 173—correct the spelling ‘through’
  31. Line 178—write ‘rays are from’ instead of ‘rays from’
  32. Line 187—write ‘were’ instead of ‘was’
  33. Kindly include the value of each bar in the figure 11
  34. Line 203—write correctly the word ‘through’ (also please check the complete manuscript for this word and correct accordingly)
  35. Line 207—write ‘are’ instead of ‘were’
  36. Line 213—remove the word ‘reasonable’

Typos, missing references and grammatical issues mentioned as detailed comments 2-17, 20-23,26-27, 29-32 and 34-36 were corrected.

Novelty part was added to the introduction section (lines 93-95).

State of art study was enhanced and novelty part was added to article (77-81 and 93-95).  

As a graphical illustration of the proposed algorithm Figures 1 and 2 were added to the article.

The disadvantages of the deep transfer learning method was discussed in the lines (95-99).

The validation of the re-training of the ResNet was presented in the Figure 7 and 8. We used only the ResNet architecture and retrained the net using two classes data sets.   

Explanation was extend (line 200-219)

The value of the each bar was included in the Fig 17.

Round 2

Reviewer 1 Report

The authors have addressed my comments.